# English colour terms carry gender and valence biases: A corpus study using word embeddings

**Domicele Jonauskaite** [1]*, **Adam Sutton**[2], **Nello Cristianini**[2], **Christine Mohr**[1]

**1** Institute of Psychology, University of Lausanne, Lausanne, Switzerland, **2** Department of Computer Science, University of Bristol, Bristol, United Kingdom

* Domicele.Jonauskaite@unil.ch

**Data Availability Statement:** The pre-existing word embeddings the authors used in this study are available at https://nlp.stanford.edu/projects/glove/. The authors do not own these word embeddings. The authors confirm that researchers

## Abstract

In Western societies, the stereotype prevails that *pink is for girls* and *blue is for boys*. A third possible gendered colour is *red*. While liked by women, it represents power, stereotypically a masculine characteristic. Empirical studies confirmed such gendered connotations when testing colour-emotion associations or colour preferences in males and females. Furthermore, empirical studies demonstrated that *pink* is a positive colour, *blue* is mainly a positive colour, and *red* is both a positive and a negative colour. Here, we assessed if the same valence and gender connotations appear in widely available written texts (Wikipedia and newswire articles). Using a word embedding method (GloVe), we extracted gender and valence biases for *blue*, *pink*, and *red*, as well as for the remaining basic colour terms from a large English-language corpus containing six billion words. We found and confirmed that *pink* was biased towards femininity and positivity, and *blue* was biased towards positivity. We found no strong gender bias for *blue*, and no strong gender or valence biases for *red*. For the remaining colour terms, we only found that *green*, *white*, and *brown* were positively biased. Our finding on *pink* shows that writers of widely available English texts use this colour term to convey femininity. This gendered communication reinforces the notion that results from research studies find their analogue in real word phenomena. Other findings were either consistent or inconsistent with results from research studies. We argue that widely available written texts have biases on their own, because they have been filtered according to context, time, and what is appropriate to be reported.

## Introduction

In Western societies, *blue* is stereotypically associated with boys and *pink* with girls [1–3]. Curiously enough, these gendered associations were initially arbitrary, but became pervasive in the early 20th century [1, 2, 4]. Nowadays, many parents continue choosing *pink* when dressing their daughters, decorating their rooms [5], or buying them toys [6]. Such an upbringing might explain why young girls show a particular liking for *pink* [7, 8]. In contrast, young adult women choose other colours as their favourite, with *blue* and *red* being the most common

can access this data in the same way they did. The authors' analyses of these word embeddings are reproducible from their own data set which are available at https://github.com/adam-sutton-1992/English-colour-terms-carry-gender-and-valence-biases-A-corpus-study-using-word-embeddings. The authors confirm they had no special access privileges.

**Funding:** This research was made possible through a Doc.CH fellowship grant to DJ (P0LAP1_175055) and a project-funding grant to CM (100014_182138) from the Swiss National Science Foundation (http://www.snf.ch/en). AS was supported by the Engineering and Physical Sciences Research Council (EP/I028153/ and EP/L016656/1; https://epsrc.ukri.org/). NC was supported by the ERC grant ThinkBIG (https://cordis.europa.eu/project/id/339365).

**Competing interests:** The authors have declared that no competing interests exist.

choices [9]. We explain these differences in colour preferences through gendered and valenced stereotypes.

Empirical studies, focused on colour preferences and colour connotations, have demonstrated that *pink* is considered to be a feminine colour and *blue* a masculine colour [3, 8, 10, 11]. *Pink* further represents groups of low social power and low social status [12–14]. Accordingly, adult women might shun *pink* to avoid being associated with these representations [9]. *Red*, on the contrary, represents being in power, dominant, and of high social status [15–18]. These representations potentially explain why adult women like *red* [9, 19–22] and why *red* carries both positive and negative connotations [23–27]. When it comes to valence, *pink* and *blue* both have been associated with mainly positive emotions [24, 27–29], although *blue* has been also associated with *sadness* [30–32].

When considering such gendered colours, we might first think about seeing them on girls' and boys' clothing, in their rooms, or their toys. Colours, however, also exist conceptually, in our minds. Colours might be gendered simply because of how we label them. If we think about *pink*, its gendered connotations might emerge not only because *pink* repeatedly occurs in feminine contexts visually, but also because it co-occurs with other feminine words in languages (e.g., *pink* is a *girly colour*). Empirical studies confirmed that colours expressed through language carry similar connotations to the same colours presented visually. Numerous studies demonstrated the red-attractiveness effect, wherein a person wearing red (man or woman) is perceived as more attractive by an opposite sex individual (for a meta-analysis, see [33]). Important here, simply mentioning that a man was wearing a red shirt had a similar effect on his attractiveness [34]. Though not focussing on the gender-loadings of colours, Jonauskaite and colleagues [35, 36] confirmed that colours, whether presented perceptually (colour patches) or semantically (colour terms), were associated with similar emotions. In the end, if the literature on embodiment and psycholinguistics holds true, an abstract meaning of a word and its respective physical representation in the world should converge on a common cognitive representation [37, 38].

We wanted to learn whether above empirical results on gendered colours can, indeed, be observed when looking at gender and valence biases in widely available written texts. Thus, we analysed gender and valence biases of colour terms in an English corpus, composed of newswire and Wikipedia articles. The latter sources both reflect and shape current standards of language use, as they are co-authored by several people and aimed at vast audiences. To extract biases, we used an artificial intelligence algorithm, focused on natural language processing (NLP). A contemporary key technique in NLP is that of word embeddings (e.g., GloVe, [39], word2vec, [40]), in which a statistical algorithm computes coordinates in a high dimensional space for each word on the basis of a reference corpus. Such NLP algorithms generally learn these coordinates from patterns of word co-occurrences in sentences. These algorithms extract not only semantic and syntactic information from everyday language, but also subtle biases in the usage of words [41]. For example, the words *nurse* or *housekeeper* frequently take a more feminine position in the semantic space than the words *pilot* or *engineer*, which take a more masculine position [41, 42].

In the current study, we used word embeddings generated from a corpus of 6 billion words by GloVe [39]. We focused on 11 basic colour terms, which included the key terms *pink*, *blue*, and *red*. Using these embeddings, we were able to score 100,000 most frequent words in the corpus in terms of similarities to the concepts of male, female, posemo, and negemo. The concepts of posemo and negemo, respectively, denote lists of positively and negatively laden words (see [43]). Words in our corpus that were closer to the list of words denoting each of the four concept [43] had higher similarity scores (also see, [44]). Afterwards, we computed gender and valence biases. We defined the gender bias as the difference between word similarities

to the concepts of male and female, and the valence bias as the difference between word similarities to the concepts of posemo and negemo. We interpreted these biases of colour terms in comparison to i) four anchor words with clear biases (i.e., *happy*, *sad*, *nun*, *priest*); and ii) a normative word population, namely the 100,000 most frequent words in the corpus. We hypothesised that the word *pink* would be biased towards femininity and positivity, while the word *blue* would be biased towards masculinity and positivity. However, *blue* might be less strongly gender-biased than *pink* [9]. We expected *red* to be embedded in both positive and negative contexts, pushing its valence bias towards zero. For its gender bias, we assumed *red* to represent power, and informed by the literature on gender stereotypes, power would represent masculinity.

## Method

We analysed gender and valence biases of 11 British English basic colour terms, namely *red*, *orange*, *yellow*, *green*, *blue*, *purple*, *pink*, *brown*, *grey*, *white*, and *black* [45, 46]. We also included four anchor words–two for the valence extremes (*happy*, *sad*) and two for the gender extremes (*priest*, *nun*). These words acted as sanity checks.

## Word embeddings

Each word within a word embedding is represented by a vector *w* of high dimensionality *d* (*d* = 300 in our study). Cosine similarity and Euclidean distances have shown the ability to represent semantic relationships, known as linear substructures, between words (e.g., GloVe, [39]). For example, vector representations for the words *man*, *woman*, *king*, and *queen* are such that:

$$king - queen \approx man - woman$$

We used a set of pre-trained word embeddings provided by GloVe [39] and available on their website (https://nlp.stanford.edu/projects/glove/). These word embeddings were trained on a corpus, formed by Wikipedia articles, downloaded in 2014, and the Gigaword5 corpus, a large archive of newswire text data collected between 1994 and 2010 [47]. This corpus contains 6 billion word-tokens in total and 400,000 unique words.

## Word similarities to concepts

We built on the method by Caliskan and colleagues [41], scoring word similarities to concepts (also see [42]). These concepts were validated by the LIWC project [43], which was created for use in social, clinical, and cognitive psychology and was based on carefully crafted and validated word lists denoting various concepts (e.g., work, family, time, etc.). We used the LIWC 2015 version [48].

In the current study, we used three LIWC word lists [48] representing four relevant concepts: i) "heshe" list, which is split to the "he" and "she" lists, ii) the "posemo" list, and iii) the "negemo" list. The "he" and "she" lists were used to establish similarities to the concepts of male and female, respectively. We acknowledge that gender is non-binary. However, we treated gender for this dataset as such, because we relied on pre-existing data, which separates gender in a binary fashion (i.e., "heshe" list of LIWC). These lists contained words like *he*, *his*, *him* (male), and *she*, *her* (female). The "posemo" list contained positively laden words like *love*, *nice*, *sweet*, while the "negemo" list contained negatively laden words like *hurt*, *ugly*, *nasty*.

These word lists were used to generate a "mean vector" (denoted $\mu$) for each concept as defined by:

$$\mu = \frac{1}{|L|} \sum_{i}^{|L|} word_i \tag{1}$$

where L is the set of all the words in a given LIWC list, and $word_i$ is the word vector representation for the $i$'th word in that list. This mean vector $\mu$ stands for the general direction in our embedding space. As we worked with four LIWC concepts, the mean vector has directions corresponding to the concepts of male, female, posemo and negemo.

We scored all words using the following cosine similarity function:

$$\langle \hat{w}, \hat{\mu} \rangle = \sum_{i=1}^{d} \frac{w_i}{\|w\|} \cdot \frac{\mu_i}{\|\mu\|} \tag{2}$$

Where $w$ is the word vector representation for a word we wish to score, $d$ is the dimension of the embedding space, and $w_i$ is the $i$'th coordinate of that word vector. Both vectors are normalised to unit length, meaning that the vector length of both $\mu$ and $w$ is 1. The result is a single real number ranging between -1 and 1. The higher the number, the more similar are the words to the concept we are scoring.

## Gender and valence biases

To calculate gender biases for each word, we subtracted similarity scores to the concept of female from those to male. Thus, a positive gender bias score indicates a bias towards masculinity and a negative score towards femininity. To calculate the valence bias for each word, we subtracted similarity scores to the concept of negemo from those to posemo. Thus, a positive valence bias score indicates that a word is biased towards positivity and a negative score towards negativity. This scoring function is defined by:

$$F(w, \mu1, \mu2) = \langle w, \mu1 \rangle - \langle w, \mu2 \rangle \tag{3}$$

For bias calculations, scores can range from -2 to 2. The extreme scores would occur when a given word is identical to one vector mean (e.g., posemo) and the opposite to another vector mean (e.g., negemo).

## Data analysis

In order to appreciate the magnitude of each bias, we compared gender and valence biases of each colour term with the distribution of biases of a normative word population. The normative word population consisted of the 100,000 most frequent words in the same corpus. We defined extreme similarities to concepts and extreme biases if the scores were below the 5th and above the 95th percentile of the normative word population.

We uploaded the code and data for these results to the following GitHub repository; https://github.com/adam-sutton-1992/English-colour-terms-carry-gender-and-valence-biases-A-corpus-study-using-word-embeddings.

## Results

### Gender bias

We first interpreted word similarities to the concepts of male and female (see Fig 1 and Table 1). Seven colour terms *red*, *blue*, *green*, *yellow*, *brown*, *white*, and *black* and the anchor

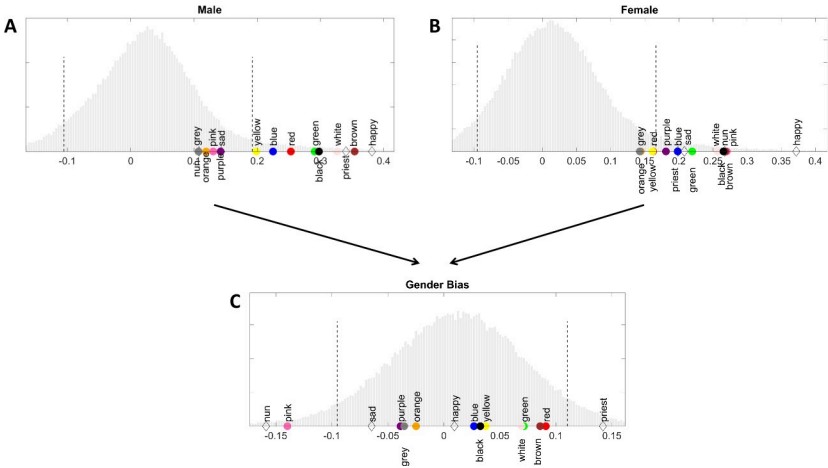

**Fig 1. Gender bias.** Distributions of similarity scores to the concepts of male (A) and female (B) well as gender biases (C) of colour terms, anchor words, and 100,000 most frequent words in our corpus. Colour terms and anchor words are marked with appropriate colours. Dashed lines indicate the 5th and 95th percentiles of the distributions (for exact values, see Table 1).

words *priest* and *happy* had higher similarities to the concept of male than 95% of the words in the normative word population (Fig 1A). Seven colour terms *blue*, *green*, *pink*, *brown*, *white*, *black*, *purple* and all anchor words had higher similarities to the concept of female than 95% of the words in the normative word population (Fig 1B). Notably, four colour terms, *green*, *blue*, *brown*, and *black*, and the anchor word *happy* had higher similarity scores to both concepts (i.e., male and female) than 95% of the words in the normative population.

When interpreting gender biases (Fig 1C), the anchor word *priest* scored higher than 95% of the words in the normative word population, meaning it was biased towards masculinity. The anchor word *nun* and the colour term *pink* scored lower than 95% of the words in the

**Table 1. Similarity scores and biases.** These are the similarity scores to the concepts of male, female, posemo, and negemo, as well as gender and valence biases of 11 colour terms and four anchor words. The position of each score in relation to the normative word population (i.e., 100,000 most frequent words) appears under "percentile". Values in bold are below the 5th or above the 95th percentile, indicating extreme similarities or biases of these words (same data as in Figs 1 and 2).

| | Male | | Female | | Gender | | Posemo | | Negemo | | Valence | |
|---|---|---|---|---|---|---|---|---|---|---|---|---|
| Word | Similarity | Percentile | Similarity | Percentile | Bias | Percentile | Similarity | Percentile | Similarity | Percentile | Bias | Percentile |
| Red | **0.2531** | 97.29 | 0.1622 | 94.71 | 0.0909 | 90.83 | **0.2659** | 96.79 | 0.1639 | 91.63 | 0.1019 | 94.83 |
| Orange | 0.1192 | 87.23 | 0.1441 | 93.06 | -0.0249 | 27.49 | 0.1326 | 89.11 | 0.0444 | 60.41 | 0.0882 | 93.04 |
| Yellow | **0.1985** | 95.31 | 0.1613 | 94.65 | 0.0372 | 66.73 | 0.1819 | 93.24 | 0.1276 | 87.50 | 0.0543 | 85.02 |
| Green | **0.2909** | 98.18 | **0.2194** | 97.55 | 0.0714 | 84.28 | **0.2652** | 96.77 | 0.0927 | 79.86 | **0.1726** | 98.83 |
| Blue | **0.2250** | 96.41 | **0.1982** | 96.74 | 0.0268 | 60.05 | **0.2690** | 96.87 | 0.0883 | 78.52 | **0.1807** | 99.01 |
| Purple | 0.1422 | 91.03 | **0.1809** | 95.88 | -0.0387 | 20.56 | 0.1819 | 93.24 | 0.1009 | 82.11 | 0.0810 | 91.80 |
| Pink | 0.1303 | 89.29 | **0.2701** | 98.79 | **-0.1397** | 1.77 | **0.2283** | 95.51 | 0.0903 | 79.17 | **0.1380** | 97.56 |
| Brown | **0.3538** | 99.10 | **0.2680** | 98.75 | 0.0858 | 89.35 | **0.2419** | 96.02 | 0.1385 | 89.00 | **0.1034** | 95.00 |
| White | 0.3261 | 98.78 | **0.2568** | 98.55 | 0.0692 | 83.42 | **0.3120** | 97.85 | 0.2007 | 94.04 | **0.1130** | 95.76 |
| Grey | 0.1072 | 84.54 | 0.1427 | 92.91 | -0.0355 | 22.15 | 0.0949 | 82.49 | 0.0213 | 48.13 | 0.0737 | 90.34 |
| Black | **0.2977** | 98.31 | **0.2652** | 98.71 | 0.0325 | 63.79 | **0.2659** | 96.79 | **0.2267** | 95.37 | 0.0392 | 79.16 |
| Nun | 0.1082 | 84.79 | **0.2670** | 98.73 | **-0.1588** | 1.22 | 0.1024 | 84.21 | 0.1143 | 85.17 | -0.0119 | 48.02 |
| Priest | **0.3405** | 98.94 | **0.1985** | 96.76 | **0.1420** | 98.47 | 0.1437 | 90.35 | 0.1665 | 91.83 | -0.0227 | 40.57 |
| Happy | **0.3812** | 99.33 | **0.3719** | 99.72 | 0.0094 | 48.59 | **0.7130** | 99.99 | **0.4568** | 99.68 | **0.2563** | 99.84 |
| Sad | 0.1433 | 91.17 | **0.2079** | 97.14 | -0.0646 | 11.17 | **0.4151** | 99.21 | **0.5475** | 99.94 | **-0.1325** | 4.17 |

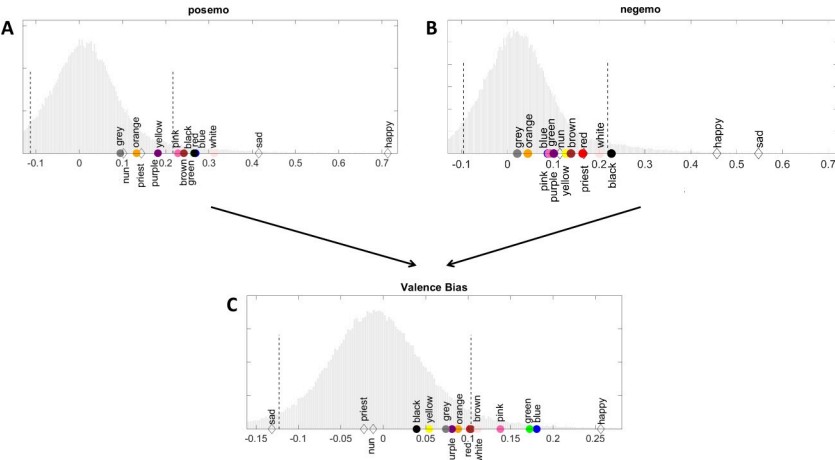

**Fig 2. Valence bias.** Distributions of similarity scores to the concepts of posemo, denoting positively laden words, (A) and negemo, denoting negatively laden words, (B) as well as valence biases (C) of colour terms, anchor words, and 100,000 most frequent words in our corpus. Colour terms and anchor words are marked with appropriate colours. Dashed lines indicate the 5th and 95th percentiles of the distributions (for exact values, see Table 1).

normative word population, meaning they were both biased towards femininity. No other colour terms or anchor words had gender bias scores outside the 5th and 95th percentiles of the normative word population, indicating they did not have extreme gender biases (see Table 1). Thus, similarities to individual concepts (i.e., male and female) do not necessarily mean that a word has a gender bias since similarities to both concepts can be positively correlated.

## Valence bias

When interpreting word similarities to the concepts of posemo and negemo (see Fig 2 and Table 1), seven colour terms, *red*, *green*, *blue*, *pink*, *brown*, *white*, and *black*, and two anchor words *happy* and *sad* had higher similarities to the concept of posemo than 95% of the words in the normative word population (Fig 2A). The colour term *black* and the anchor words *happy* and *sad* had higher similarities to the concept of negemo than 95% of the words in the normative word population (Fig 2B). Notably, *black*, *happy*, and *sad* had higher similarity scores to both concepts (i.e., posemo and negemo) than 95% of the words in the normative population.

When interpreting valence biases (Fig 2C), five colour terms, *green*, *blue*, *pink*, *brown*, and *white*, and the anchor word *happy* scored higher than 95% of the words in the normative word population, meaning they were biased towards positivity. The anchor word *sad* scored lower than 95% of the words in the normative word population, meaning it was biased towards negativity. No other colour terms or anchor words had valence bias scores outside the 5th and 95th percentiles of the normative word population, indicating they did not have extreme valence biases (see Table 1). Thus, similarities to individual concepts (i.e., posemo and negemo) do not necessarily mean that a word has a valence bias since similarities to both concepts can be positively correlated.

## Discussion

We investigated whether empirical results on gender and valence biases for colour terms are reflected in an English corpus composed of newswire and Wikipedia articles. Based on previous empirical studies, we expected *pink* to be biased towards femininity and positivity, and

*blue* towards masculinity and positivity. We expected *red* to be biased towards masculinity, while having no particular valence bias because *red* carries both positive and negative connotations [23–27]. To this end, we scored embeddings of the colour terms *pink*, *blue*, and *red*, which we obtained from GloVe [39]. Scoring was done in terms of similarities to the concepts of male, female, posemo and negemo, that means positively and negatively laden words, as defined by the LIWC project [43]. We then computed gender and valence biases as differences between similarities to the concepts of male vs. female, and posemo vs. negemo, respectively. We did the same for the remaining eight basic colour terms, four anchor words (i.e., *happy*, *sad*, *nun*, *priest*), and the 100,000 most frequent words in the corpus.

First, we checked for extreme similarities of our colour terms to the concepts of male and female. We defined extreme word similarities when a colour term had a higher similarity with a concept than 95% of the 100,000 most frequent words in the corpus (i.e., normative word population). These comparisons revealed that i) *red* and *blue* were closer to the concept of male and ii) *blue* and *pink* were closer to the concept of female than 95% of words in the normative word population. That meant that *blue* was close to both concepts, yielding overall no gender bias. When looking at gender biases in comparison to the normative word population, *pink* was the only colour term with a gender bias. We confirmed its bias towards femininity. Against our predictions, neither *blue* nor *red* were biased towards masculinity and no other colour term had a strong gender bias. Hence, we concluded that only *pink* conveys gender biased information (femininity) in these texts.

Second, we checked for extreme similarities of our colour terms to the concepts of posemo and negemo. Comparisons with the normative word population revealed that colour terms *red*, *blue*, and *pink* were closer to the concept of posemo, while no colour terms of interest were closer to the concept of negemo than 95% of the most frequent words in the normative word population. When looking at valence biases, as predicted, *pink* and *blue* were biased towards positivity and *red* did not have a strong valence bias. Other positively biased colour terms were *green*, *white*, and *brown*, while no colour term was negatively biased.

The femininity bias of *pink* mirrors previous empirical findings [3, 10, 11, 49, 50]. In contrast, findings on *blue* did not confirm a masculinity bias in our corpus, unlike previous empirical studies would have suggested [1, 3, 10, 11]. Indeed, *blue* seems gender-neutral, equally liked by males and females of all ages [9, 51–56]. *Blue* might turn into a symbol of masculinity, or boyhood, only when paired with *pink*, which in turn is a symbol of femininity and girliness, because *pink* is avoided by boys, men, and some adult women (see a more in-depth reasoning in [9, 57]). In fact, a recent study using a Stroop paradigm demonstrated that masculine words written in pink ink were perceived as being more incongruent than feminine words written in blue ink [57]. As for *red*, we expected, but did not find a masculinity bias, due to its associations with power and dominance [15–18]. Indeed, *red* is favoured by many women and might be a symbol of femininity [9, 19–22]. Thus, one can argue that *red* represents both masculine and feminine characteristics resulting in a negligible bias in our dataset.

Findings on valence biases were also congruent with previous empirical results. Positivity biases of *pink* and *blue* were compatible with numerous previous studies [15, 24, 27–29, 35, 36], even though *blue* also carries some negative connotations. Among English speakers, *blue* is associated with *sadness* in addition to several positive connotations [27, 30–32]. We did not expect, and did not observe, that *red* had a strong positivity or negativity bias. In empirical studies, *red* has been linked to positive emotions like *love*, *joy*, and *pleasure* as well as negative emotions like *anger* and *hate* [24, 26, 27, 35, 36, 58, 59].

Beyond our interest in (potentially) gendered colours, we confirmed only some of the previously observed valence biases for the remaining basic colour terms. Positivity biases of *green*, and *white* were compatible with previous studies [24, 25, 27, 32, 60]. However, we did not

observe a positivity bias of *yellow*, despite numerous studies reporting associations with *joy*, *happiness*, and other positive emotions [24, 27, 29, 35, 59, 61, 62]. Unexpectedly, we observed a positivity bias of the colour term *brown*. In nearly all previous empirical studies, *brown* carried negative associations, including associations with *disgust* and *boredom* [27, 30, 35, 59]. The association with *disgust* is likely due to evolutionary important experiences, like rotten food and faeces, supposedly explaining why people do not like *brown* [52]. Experiences of rotten food and faeces might, however, be taboo subjects in widely available newswires and Wikipedia articles. Rather, people might be mentioning *brown* in the contexts of coffee and chocolate, both of which are positive experiences for most. Finally, we did not observe negativity biases of *black* and *grey*, despite previous studies showing that black is associated with *sadness*, *fear*, and *death* and *grey* with *sadness* and *disappointment* [24, 26, 27, 58, 60].

Our approach to detect biases in written text is methodologically different from former research studies. Maybe, these differences can explain some discrepancies between the current and previous studies. We extracted biases in an English corpus, while previous research usually tested for implicit or explicit colour meaning via individual ratings, associations, or other questioning (e.g., [3, 8, 10, 11, 15, 23–28, 60]). Additionally, when we consider the posemo and negemo lists, their names (positive emotions, negative emotions) might give rise to the idea that these words represent emotions, but most words do not describe emotions *per se*, but positively or negatively laden concepts (e.g., see definitions of emotions in [63–66]). For instance, *sadness*, *anger*, or *joy* would represent emotions, but not *nice*, *kiss*, or *sceptical*. Differences in definitions of concepts, study material and procedures might lead to varied study results.

## Limitations and future directions

In the field of engineering, researchers use word embeddings as input for downstream artificial intelligence tasks. Wikipedia and newswire articles are considered standard sources for learning word embeddings. One could argue that such widely available text sources reflect current standards of language use, and by inference, current thinking, as they are authored by several people and aimed at vast audiences. However, our method has at least two limitations. First, in our case, the text sources were in English, likely written by native and non-native speakers. Therefore, our results should not be generalised to other languages and populations. Second, the method leaves us without socio-demographic information about the writers (e.g., their gender, age, country) or their personalities (but see [67]). In the current design, we cannot know if the reported biases are common to everyone or are rather representative for only a particular part of society or a single culture. To obtain such information, we would need controlled studies. In particular, we would have to be able to link personal, socio-demographic, linguistic, and geographical information to an individual and their behavioural output (see also[27, 68–70]).

Worth noting, neither our corpus study nor empirical studies reflect spontaneous conversations. When setting up research studies, researchers decide beforehand what they wish to test, designing the method accordingly. In the current and other corpus studies, one extracts meaning from large corpora. In our case, we used newswire and Wikipedia articles, conveying information that was likely filtered by topic, and presented in a socially and politically acceptable manner. If we take the terms *black* and *green*, the former might dominate discussions around racial and the latter around environmental issues. Furthermore, obvious descriptors might not be mentioned. A writer might not see the necessity to state that *grass is green*, or *faeces are brown*. Finally, the contents of corpora might depend on when and where they were published, as topics vary in popularity over time and between places. Most recently, gender issues were again intensively discussed with the prominence of the #MeToo movement.

To understand how colour terms are used in spontaneous language, studies could analyse written communications on social media like Twitter (see an example in [71]) or conduct observational studies. In social media, communications might be more spontaneous, and thus closer to what people think and feel. Yet, these communications have other drawbacks, such as being limited in length, or favouring omissions and incomplete sentences.

## Conclusions

We investigated whether *pink*, *blue*, and *red* have gender and valence biases in a written text corpus. With the help of artificial intelligence technology, we could show that *pink* was the only gendered colour, biased towards femininity. These results show that artificial intelligence can be used to assess how empirical results, often from the laboratory research studies, may relate to how people use colour terms in written texts. For some colours, our corpus study mirrored empirical study results (i.e., *pink* and femininity, *blue* and *pink* and positivity), for others, we observed differences (e.g., *brown* and positivity). Thus, we argued that written texts not only reflect human thought processes, but yield biases on their own, potentially due to selection in reporting.

## Author Contributions

**Conceptualization:** Domicele Jonauskaite, Nello Cristianini, Christine Mohr.

**Data curation:** Adam Sutton, Nello Cristianini.

**Formal analysis:** Adam Sutton, Nello Cristianini.

**Supervision:** Nello Cristianini, Christine Mohr.

**Visualization:** Domicele Jonauskaite.

**Writing – original draft:** Domicele Jonauskaite, Adam Sutton, Nello Cristianini, Christine Mohr.

**Writing – review & editing:** Domicele Jonauskaite, Adam Sutton, Nello Cristianini, Christine Mohr.

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
