## [Decision Letter · Decision Letter 0]

1 Oct 2020

PONE-D-20-26001

Colour terms carry gender and valence biases in natural language corpora

PLOS ONE

Dear Dr. Jonauskaite,

Thank you for submitting your manuscript to PLOS ONE. After careful consideration, we feel that it has merit but does not fully meet PLOS ONE’s publication criteria as it currently stands. Therefore, we invite you to submit a revised version of the manuscript that addresses the points raised during the review process.

Both of the reviewers provide numerous valuable suggestions. Please take them into account to every possible extent.

We look forward to receiving your revised manuscript.

Kind regards,

Søren Wichmann, PhD

Academic Editor

PLOS ONE

Journal Requirements:

Reviewers' comments:

Reviewer's Responses to Questions

**Comments to the Author**

1. Is the manuscript technically sound, and do the data support the conclusions?

Reviewer #1: Yes

Reviewer #2: Partly

2. Has the statistical analysis been performed appropriately and rigorously? 

Reviewer #1: Yes

Reviewer #2: Yes

3. Have the authors made all data underlying the findings in their manuscript fully available?

Reviewer #1: No

Reviewer #2: Yes

4. Is the manuscript presented in an intelligible fashion and written in standard English?

Reviewer #1: Yes

Reviewer #2: Yes

5. Review Comments to the Author

Reviewer #1: The reviewer self-identifies as Bodo Winter and is available for follow-up questions. Please see the attached PDF for detailed feedback, which can also be accessed under this link:

http://appliedstatisticsforlinguists.org/PLOS_ONE_gender.pdf

Reviewer #2: This article presents an analysis of the gender and valence bias of color terms through a Glove embeddings model fitted against Wikipedia and news. The authors find results that are consistent with previous results from experiments.

The methodology is appropriate and the presentation is good. However, I have some comments regarding the interpretation of results and some methodological details:

- Results are framed as "in natural language" in general, but are only based on a combination of Wikipedia and news. The authors fairly mention this in their limitations section, but to generalize results to natural language it would be necessary to replicate the results with other corpora and/or languages. As there are pre-trained embeddings for many corpora and LIWC present in a wide variety of languages, it would not be hard for the authors to add replications that support their generalization to natural language. In case this is not possible, the abstract and conclusions of the article can be further contextualized to point that results only apply to one English corpus, without generalizing to natural language.

- Positive and Negative Affect in LIWC are treated as ends of a single dimension of valence, while in LIWC they are conceived as two variables that can co-occur, not just in text but also at the individual level. Even if they are not completely orthogonal, mapping valence as the subspace between PA and NA in the embeddings space induces a situation in which the middle point is not well defined. In the definition of valence in the paper, a word that has the same cosine similarity to the centroid of PA words as to the centroid of NA words is neutral, but this does not have to be the case. If the authors really want to analyze emotional valence, there is a large number of affective norms lexica that would allow them to approximate a mapping from the embeddings space to the dimension of valence as defined in those lexica. If they wanted to study something more general in the lines of evaluative meanings, other lexica like the General Inquirer would be more suitable than LIWC. I recommend the authors to revise the assumption of their calculation of valence bias and revise whether zero in their scale really means a neutral bias.

- The null model used for statistical analysis assumes that gender and valence bias are orthogonal. From previous works in sentiment analysis, we know that this is unlikely to be the case in many models like Glove. This assumed orthogonality can be seen in the Figure, where the limits for significance describe a square. Before concluding what associations are significant and which are not, the authors can use the bivariate distribution of valence and gender bias in their all-pairs sample and test each 2-D point of colors against that more realistic null model. This might very well add power to the method.

- The statistical test assumes normality of the distribution under the null, but the SI figure seems to show quite some deviation from a normal distribution, especially for the case of valence bias (if the lines in the figure are indeed a normal fit, which is not explained). I think the authors do not need this normal assumption at all, they can just use the quantiles of the bivariate distribution composed of the joint values shown in both SI figures, labelling for example which points are beyond the 95% quantiles in their color word analysis.

- On a clarity note, authors should report which version of LIWC they used. They cite the 2001 version, which would be quite outdated compared to the 2015 version. They should also explain how they mapped entries in the LIWC dictionary, especially those with wildcards or other rules, to individual words in the embeddings model.

6. PLOS authors have the option to publish the peer review history of their article (what does this mean?). If published, this will include your full peer review and any attached files.

Reviewer #1: **Yes: **Bodo Winter

Reviewer #2: No

---

## [Author Response · Author response to Decision Letter 0]

31 Mar 2021

Our response to reviewers has been attached separately.

---

## [Decision Letter · Decision Letter 1]

12 Apr 2021

PONE-D-20-26001R1

English colour terms carry gender and valence biases: A corpus study using word embeddings

PLOS ONE

Dear Dr. Jonauskaite,

Thank you for submitting your revised manuscript to PLOS ONE. You can essentially regard this as accepted, but Reviewer #1 mentions a "tiny thing" that would still merit a bit of revision.

A rebuttal letter that responds to each point raised by the academic editor and reviewer(s). You should upload this letter as a separate file labeled 'Response to Reviewers'. No need to be wordy here.A marked-up copy of your manuscript that highlights changes made to the original version. You should upload this as a separate file labeled 'Revised Manuscript with Track Changes'.An unmarked version of your revised paper without tracked changes. You should upload this as a separate file labeled 'Manuscript'.

We look forward to receiving your revised manuscript.

Kind regards,

Søren Wichmann, PhD

Academic Editor

PLOS ONE

Journal Requirements:

Reviewers' comments:

Reviewer's Responses to Questions

**Comments to the Author**

1. If the authors have adequately addressed your comments raised in a previous round of review and you feel that this manuscript is now acceptable for publication, you may indicate that here to bypass the “Comments to the Author” section, enter your conflict of interest statement in the “Confidential to Editor” section, and submit your "Accept" recommendation.

Reviewer #1: All comments have been addressed

2. Is the manuscript technically sound, and do the data support the conclusions?

Reviewer #1: Yes

3. Has the statistical analysis been performed appropriately and rigorously? 

Reviewer #1: Yes

4. Have the authors made all data underlying the findings in their manuscript fully available?

Reviewer #1: Yes

5. Is the manuscript presented in an intelligible fashion and written in standard English?

Reviewer #1: Yes

6. Review Comments to the Author

Reviewer #1: Excellent job at addressing my comments from the first round. I also really appreciated the justification for those aspects where the authors stood their ground and argued (in my mind very well) why it makes not that much sense to implement certain changes. I like the new limitations section and the manuscript reads very well overall. I think this is a straightforward accept.

I'd just change one tiny thing: You say: "This should be the case according to theories in cognitive sciences [37,38]." But "Theories in cognitive science" could be anything or nothing. It's exceedingly vague and I think you can be more specific here about what sort of theories these are? As there is a lot of theoretical diversity WITHIN cognitive science, I could easily see some cognitive scientists been thrown off by such a broad sweep statement. Anyway, this is all but a minor fix.

Thank you for submitting a great paper and also for your thoughtful reviewer response.

7. PLOS authors have the option to publish the peer review history of their article (what does this mean?). If published, this will include your full peer review and any attached files.

Reviewer #1: **Yes: **Bodo Winter

---

## [Author Response · Author response to Decision Letter 1]

28 Apr 2021

We have adapted the problematic sentence in the introduction as suggested by Dr Winter.

---

## [Editor Report · Decision Letter 2]

29 Apr 2021

English colour terms carry gender and valence biases: A corpus study using word embeddings

PONE-D-20-26001R2

Dear Dr. Jonauskaite,

We’re pleased to inform you that your manuscript has been judged scientifically suitable for publication and will be formally accepted for publication once it meets all outstanding technical requirements.

Note there is a typo in added sentence "if the literature on embodiment and psycholinguistics hold true, " (hold -> holds). You can correct that in the proofs.

Kind regards,

Søren Wichmann, PhD

Academic Editor

PLOS ONE
---

## [Editor Report · Acceptance letter]

21 May 2021

PONE-D-20-26001R2 

English colour terms carry gender and valence biases: A corpus study using word embeddings 

Dear Dr. Jonauskaite:

I'm pleased to inform you that your manuscript has been deemed suitable for publication in PLOS ONE. Congratulations! Your manuscript is now with our production department. 

Kind regards, 

on behalf of

Dr. Søren Wichmann 

Academic Editor

PLOS ONE